# Lack of Electron Acceptors Contributes to Redox Stress and Growth Arrest in Asparagine-Starved Sarcoma Cells

**DOI:** 10.3390/cancers13030412

**Published:** 2021-01-22

**Authors:** Christoph Bauer, Meret Quante, Willemijn B. Breunis, Carla Regina, Michaela Schneider, Geoffroy Andrieux, Oliver Gorka, Olaf Groß, Melanie Boerries, Bernd Kammerer, Simone Hettmer

**Affiliations:** 1Division of Pediatric Hematology and Oncology, Department of Pediatric and Adolescent Medicine, University Medical Center Freiburg, University of Freiburg, Mathildenstrasse 1, 79106 Freiburg, Germany; christoph.bauer@uniklinik-freiburg.de (C.B.); meret.quante@uniklinik-freiburg.de (M.Q.); carla.regina@uniklinik-freiburg.de (C.R.); michaela.schneider@uniklinik-freiburg.de (M.S.); 2Center for Biological Systems Analysis (ZBSA), University of Freiburg, Habsburgerstrasse 49, 79104 Freiburg, Germany; 3Faculty of Biology, University of Freiburg, Schänzlestrasse 1, 79104 Freiburg, Germany; 4Department of Oncology and Children’s Research Center, University Children’s Hospital, Steinwiessstrasse 75, 8032 Zürich, Switzerland; Willemijn.Breunis@kispi.uzh.ch; 5Institute of Medical Bioinformatics and Systems Medicine, Faculty of Medicine, Medical Center-University of Freiburg, University of Freiburg, 79104 Freiburg, Germany; geoffroy.andrieux@uniklinik-freiburg.de (G.A.); melanie.boerries@uniklinik-freiburg.de (M.B.); 6German Cancer Consortium (DKTK), Freiburg, Germany and German Cancer Research Center (DKFZ), Im Neuenheimer Feld 280, 69120 Heidelberg, Germany; 7Institute of Neuropathology, Faculty of Medicine, University of Freiburg, Breisacher Strasse 64, 79106 Freiburg, Germany; oliver.gorka@uniklinik-freiburg.de (O.G.); olaf.gross@uniklinik-freiburg.de (O.G.); 8Signaling Research Center BIOSS and CIBSS, University of Freiburg, Schänzlestrasse 18, 79104 Freiburg, Germany; 9Center for Basics in NeuroModulation (NeuroModulBasics), Faculty of Medicine, University of Freiburg, Breisacher Strasse 64, 79106 Freiburg, Germany; 10Comprehensive Cancer Centre Freiburg (CCCF), Medical Center-University of Freiburg, Hugstetter Strasse 49, 79106 Freiburg, Germany; 11Spemann Graduate School of Biology and Medicine (SGBM), Albertstraße 19A, 79104 Freiburg, Germany

**Keywords:** sarcoma, metabolomics, asparagine starvation, reductive stress

## Abstract

**Simple Summary:**

Cancer cells require amino acids to grow and survive. Amino acid starvation inhibits cancer growth. This study investigates why the non-essential amino acid asparagine is important for cancer growth by examining metabolite composition in asparagine-deprived sarcoma cells compared to control cells with normal asparagine access. Our experiments show that asparagine deprivation results in an imbalance between certain antioxidants and free radicals in the cell. Chemicals which impair the regeneration of antioxidants in the cell augment the growth inhibition caused by asparagine starvation.

**Abstract:**

Amino acids are integral components of cancer metabolism. The non-essential amino acid asparagine supports the growth and survival of various cancer cell types. Here, different mass spectrometry approaches were employed to identify lower aspartate levels, higher aspartate/glutamine ratios and lower tricarboxylic acid (TCA) cycle metabolite levels in asparagine-deprived sarcoma cells. Reduced nicotinamide adenine dinucleotide (NAD^+^)/nicotinamide adenine dinucleotide hydride (NADH) ratios were consistent with redirection of TCA cycle flux and relative electron acceptor deficiency. Elevated lactate/pyruvate ratios may be due to compensatory NAD^+^ regeneration through increased pyruvate to lactate conversion by lactate dehydrogenase. Supplementation with exogenous pyruvate, which serves as an electron acceptor, restored aspartate levels, NAD^+^/NADH ratios, lactate/pyruvate ratios and cell growth in asparagine-deprived cells. Chemicals disrupting NAD^+^ regeneration in the electron transport chain further enhanced the anti-proliferative and pro-apoptotic effects of asparagine depletion. We speculate that reductive stress may be a major contributor to the growth arrest observed in asparagine-starved cells.

## 1. Introduction

Amino acids are integral components of cancer cell metabolism. They serve as building blocks for peptide synthesis, as an energy source and as precursor molecules to secure an adequate supply of nucleotides, neurotransmitters and nitrogen. They also contribute to redox balance and serve as substrates for post-translational and epigenetic modifications [1]. In recent years, there has been increasing interest in the role of essential and non-essential amino acids in the metabolic equilibrium that enables tumor cell proliferation. Despite the fact that glutamine is a non-essential amino acid, many cancer cells were shown to be addicted to glutamine, which fuels the tricarboxylic acid (TCA) cycle and contributes to the synthesis of lipids, nucleotides and non-essential amino acids [2]. Published studies by ourselves and others also demonstrated that the growth and survival of various types of cancer cells, including sarcoma cells, depend on adequate availability of the non-essential amino acid asparagine (Asn) [3,4].

Mammalian cells maintain intracellular asparagine availability through import from the extracellular space [5] and de novo synthesis from glutamine (Gln) and aspartate (Asp) through a unidirectional ATP-dependent reaction catalyzed by asparagine synthetase (ASNS) (Figure 1) [6]. Physiological asparagine concentrations are 5.1 mg/L (38.8 µM [7]) and 10.9 mg/L (82.4 µM [8]) in mouse and human plasma, respectively (Figure 2a,b). Asparagine uptake from the extracellular space and cell-intrinsic asparagine synthesis secure intracellular asparagine concentrations below 132 ng/L (1 nM) [5,6].

ASNS expression in various types of solid tumor cells correlates with higher tumor grade, a propensity to metastasize and poor patient survival [9,10]. If cells are deprived of nutrients, including asparagine, a conserved transcriptional program known as the integrated stress response is activated to restore homeostasis through upregulation of various nutrient transporters and enzymes, including ASNS [4,6]. In other words, nutrient-deprived cells consume nitrogen and ATP to maintain intracellular availability of asparagine. Known functions of asparagine in tumor cells include the translation of new peptides [3], activation of mechanistic Target of Rapamycin complex 1 (mTORC1) signaling [4,5] and use as an amino acid exchange factor to regulate uptake of other amino acids from the extracellular space [5].

Asparagine availability in cancer cells serves as a therapeutic target. Asparagine depletion through treatment with bacterially derived asparaginase has long been established as an important strategy in the treatment of leukemias expressing low levels of ASNS [6]. Asparaginase was also shown to reduce the in vitro growth of sarcoma cells. Genetic silencing of ASNS combined with depletion of systemic asparagine via asparaginase decreased sarcoma growth in vivo [3]. Yet, sarcoma cells express high levels of ASNS and therefore rely less on environmental asparagine supply. Indeed, asparaginase sensitivity of sarcoma cells is moderate to poor when compared to lymphoblasts [3,11].

In this study, we interrogated changes in the sarcoma metabolome induced by asparagine depletion to better understand why cancer cells depend on adequate asparagine availability and to identify chemically actionable vulnerabilities that may be exploited to potentiate asparaginase effects. Our studies revealed relative excess of reducing equivalents in asparagine-starved sarcoma cells. We also report synergistic effects of asparaginase and complex 1 inhibitors, which block regeneration of nicotinamide adenine dinucleotide (NAD^+^) in the electron transport chain and enhance reductive stress in asparagine-starved sarcoma cells.

## 2. Results

### 2.1. Asparagine Deprivation of Mouse Sarcoma Cells

Intracellular asparagine was depleted in *KRAS*-driven mouse sarcoma cells [12] by abrogating asparagine synthesis via short hairpin RNA (shRNA)-mediated ASNS knockdown and culture in asparagine-free medium (A-N0). Intracellular asparagine content in A-N0 cells was reduced to <66% compared to asparagine content in shLuc (C-N5) and wild-type (W-N5) sarcoma cells grown in medium with 5 mg/L asparagine (Figure 2c). Relative asparagine content in shASNS, shLuc and wild-type cells grown in medium with 0 mg/L asparagine, 5 mg/L asparagine and 100 mg/L asparagine was compared by LC-MS. Supplementation with 5 mg/mL asparagine did not reverse the effects of the shASNS knockdown. Supplementation with 100 mg/mL asparagine raised asparagine content in shASNS, shLuc and wild-type cells to supraphysiological levels (Appendix A). There was minimal ASNS expression in shASNS cells cultured in medium containing near-physiological asparagine concentrations at 5 mg/L (38 µM) or excess asparagine concentrations at 100 mg/L (757 µM) (Figure 2d). By contrast, shASNS cells cultured in asparagine-free medium expressed slightly higher levels of ASNS compared to shASNS cells grown in medium containing physiological or excess asparagine (Figure 2d). Increased ASNS expression in shASNS cells grown in asparagine-free medium was more pronounced after 24 h than after 6 h in asparagine-free conditions (Figure 2d). Compensatory upregulation of ASNS expression in asparagine-low conditions is consistent with previously described activating transcription factor 4 (ATF4)-mediated upregulation of ASNS in asparagine-deprived cells due to cellular programs aimed at restoring nutrient homeostasis [6]. Increased processing of LC3I/II indicated increased autophagocytosis in shASNS cells compared to wild-type (WT) cells after culture in asparagine-free medium for 24 h and may also contribute to restoring homeostasis. Increased autophagocytosis was not yet observed after culture in asparagine-free medium for 6 h (Figure 2e).

### 2.2. Asparagine Starvation Impacts Proliferation, Apoptosis and Autophagy of Mouse Sarcoma Cells

Asparagine depletion strongly impeded *KRAS*-driven mouse sarcoma cell growth (Figure 2f). When cells were cultured in asparagine-free medium, the growth of shASNS sarcoma cells (marked by the red line) was significantly reduced compared to the growth of wild-type (WT, marked by the blue line) or shLuc control cells (marked by the gray line, *p* < 0.0001, Figure 2f). Furthermore, in medium containing 5 mg/L asparagine, the growth of shASNS cells was reduced compared to WT cells (*p* < 0.0001, Figure 2g). However, the reduction in growth was less pronounced in shASNS cells grown in medium containing 5 mg/L asparagine compared to those cultured in asparagine-free medium (Figure 2f–g). In medium containing excess asparagine, shASNS, shLuc and WT sarcoma cells grew equally well (Figure 2h). This is consistent with previous observations, which demonstrate that sarcoma growth depended on sufficient asparagine availability [3].

Physiological glucose concentrations are 1.5 g/L (8.3 mM) in mouse and 0.97 g/L (5.4 mM) in human plasma (Figure 2a,b). Physiological glutamine concentrations are 63.0 (0.43 mM) and 85.6 mg/L (0.59 mM) in mouse and human plasma, respectively [7] [8] (Figure 2a,b). High-glucose Dulbecco’s Modified Eagle’s Medium (DMEM) contains 4.5 g/L (25.0 mM) glucose, 584.6 mg/L (4 mM) glutamine and no asparagine. The effects of asparagine deprivation by shASNS knockdown in asparagine-free conditions were investigated in medium containing different glutamine and glucose concentrations (Appendix A). Asparagine depletion reduced the proliferation of shASNS mouse sarcoma cells (A) cultured in medium containing low glucose (0.5 g/L)/physiological glutamine (73.1 mg/L) (Appendix A, 4, *p* < 0.0001), high glucose (4.5 g/L)/physiological glutamine (73.1 mg/L) (Appendix A, 3, *p* < 0.0001) and high glucose (4.5 g/L)/high glutamine (584.6 mg/L) (Appendix A, 2, *p* < 0.0001) levels. Similarly, asparagine depletion raised the percentage Annexin V^+^/7AAD^-^ apoptotic cells in low glucose (0.5 g/L)/physiological glutamine (73.1mg/L) (Appendix A, 4, *p* < 0.01), high glucose (4.5 g/L)/physiological glutamine (73.1 mg/L) (Appendix A, 3, *p* < 0.01) and high glucose (4.5 g/L)/high glutamine (584.6 mg/L) (Appendix A, 2, *p* < 0.0001) conditions (Appendix A). Finally, asparagine depletion increased conversion of LC3I to LC3II in low/high glucose and physiological/high glutamine conditions in shASNS (A) mouse sarcoma cells compared to wild-type (W) or shLuc (C) control cells (Appendix A). These changes in proliferation, apoptosis and autophagy of asparagine-depleted cells were reversed by exogenous supplementation with 100 mg/L asparagine (Appendix A). Culture in asparagine-free medium did not impede proliferation (Appendix A) or increase apoptosis (Appendix A, 5–7) of shLuc (C) control cells in low/high glucose and physiological/supraphysiological glutamine conditions.

In medium with low glutamine content (7.3 mg/L), both shASNS cells (A) and shLuc control (C) cells exhibited low proliferative activity (Appendix A, 8–9) and contained a higher proportion of Annexin V^+^/7AAD^−^ apoptotic cells (Appendix A, 8–9 and Appendix A). Supplemental asparagine partially rescued the effects of glutamine deprivation on apoptosis (Appendix A, 8, *p* < 0.05) but not on proliferation (Appendix A, 8, ns). This is consistent with published studies demonstrating the reversal of glutamine-depletion-induced apoptosis by exogenous asparagine in tumor cells [10]. We did not observe any effects of asparagine starvation on LC3I-to-LC3II conversion in low glutamine conditions (Appendix A).

### 2.3. Metabolic Adaptation of Asparagine-Starved Mouse Sarcoma Cells

To investigate the effects of asparagine availability on the metabolome of *KRAS*-driven mouse sarcoma cells, asparagine availability was modified by exposing shASNS sarcoma cells (A), shLuc control sarcoma cells (C) and wild-type sarcoma cells (W) to asparagine-free medium or medium containing near-physiological asparagine concentrations (5 mg/L) or supraphysiological asparagine concentrations (100 mg/L) for 2 days (Figure 3a). Glucose and glutamine were supplied at supraphysiological concentrations (4.5 g/L and 584.6 mg/L, respectively). Cell lysates were evaluated by gas chromatography–mass spectrometry (GC-MS).

Principal component analyses highlighted global differences in the metabolome of shASNS cells cultured with 0 (2-A-N0; marked in red), 5 (2-A-N5; marked in yellow) or 100 mg/L (2-A-N100; marked in green) asparagine as opposed to the metabolome of shLuc cells grown with 0 (3-C-N0), 5 (3-C-N5) or 100 mg/L (3-C-N100) asparagine or wild-type cells grown with 0 (1-W-N0), 5 (1/2/3-W-N5) or 100 mg/L (1-W-N100) asparagine (all marked in shades of blue). Importantly, shASNS cells supplemented with excess asparagine at 100 mg/L (2-A-N100, marked in green) clustered with shLuc and WT cells, suggesting that exogenous supplementation with asparagine reversed the metabolomic changes induced by asparagine deprivation (Figure 3b, Appendix A).

### 2.4. Redirection of TCA Cycle flux in Asparagine-Depleted Mouse Sarcoma Cells

The global metabolome of shASNS sarcoma cells grown without supplemental asparagine (A-N0) was marked by low aspartate levels (Figure 3c) compared to that of shASNS cells grown in medium with excess asparagine (A-N100) or control shLuc or WT cells grown in medium with physiological asparagine content (C-N5, W-N5). Aspartate provides the carbon backbone for de novo synthesis of asparagine in a unidirectional ATP-dependent aminotransferase reaction [6]. Aspartate transport into most mammalian cells is inefficient. Thus, aspartate supply (Figure 1, marked in yellow) ultimately depends on conversion of glutamine to glutamate to alpha-ketoglutarate, which enters the TCA cycle and is converted into aspartate via reductive carboxylation (Figure 1, marked in dark gray) or oxidative decarboxylation (Figure 1, marked in light gray) [13,14,15] (Figure 1). Indeed, lower aspartate (Figure 3c), lower glutamate (Figure 3d) and lower glutamine (Figure 3e) levels were accompanied by an increase in aspartate/glutamine ratios (Figure 3f) in asparagine-starved A-N0 compared to control W-N5 and C-N5 cells. These observations suggest that glutamine is shunted towards aspartate synthesis through redirection of tricarboxylic acid (TCA) cycle flux (Figure 1). This was further supported by reduced citrate (Figure 3g), reduced alpha-ketoglutarate (α-KG) (Figure 3h), reduced malate (Figure 3i) and reduced oxaloacetate (Figure 3j) levels in asparagine-starved A-N0 compared to control W-N5 and C-N5 cells. Supplementation with excess asparagine in A-N100 partially reverted the intracellular content of these metabolites in shASNS cells to levels similar to those in control W-N5 and C-N5 cells (Figure 3c–j).

Transcriptional changes in asparagine-depleted cells were evaluated by RNA-Seq of shASNS cells grown in medium with 0 mg/L asparagine (A-N0), shASNS cells in medium with 5 mg/L asparagine (A-N5), shASNS cells in medium with 100mg/L asparagine (A-N100) and shLuc cells in medium with 5 mg/L asparagine (C-N5). A pathway analysis comparing A-N0 and A-N5 to A-N100 and C-N5 using the Kyoto Encyclopedia of Genes and Genomes (KEGG) database demonstrated that the top three pathways enriched among transcripts upregulated in asparagine-depleted A-N0 and A-N5 cells compared to A-N100 and C-N5 cells (Appendix A) included ribosome (Appendix A), aminoacyl transfer RNA (tRNA) biosynthesis (Appendix A) and oxidative phosphorylation (Figure 3k) (FDR < 0.05, Appendix A). Differentially regulated genes involved in oxidative phosphorylation included the succinate dehydrogenase subunit *Sdhb* (logFC 0.24, Appendix A, Figure 3k) and the alpha-ketoglutarate dehydrogenase subunit *Ogdhl* (logFC 0.79, Appendix A). Higher *Ogdhl* and *Sdhb* transcript levels in asparagine-starved A-N0 compared to A-N100 and control C-N5 cells were confirmed by qPCR (Figure 3l–m). These observations further support altered TCA cycle flux in asparagine-starved sarcoma cells.

### 2.5. Relative Lack of the Electron Acceptor NAD^+^ in Asparagine-Starved Cells

Oxidative phosphorylation involves a continuous flow of electrons from electron donors to electron acceptors. For example, NAD^+^ serves as an electron carrier in a continuous cycle of reduction to nicotinamide adenine dinucleotide hydride (NADH) (Figure 1, marked in red) and oxidation back to NAD^+^ (Figure 1, marked in green). Liquid chromatography–mass spectrometry (LC-MS, Figure 4a) was used to determine NAD^+^ and NADH levels in asparagine-deprived and control cells. Notably, NAD^+^/NADH ratios were significantly reduced in asparagine-starved A-N0 cells grown without supplemental asparagine (A-N0) compared to shLuc (C-N5, *p* < 0.001) and WT cells (W-N5, *p* < 0.01) cultured in near-physiological asparagine conditions, consistent with the relative lack of electron acceptors in asparagine-deprived cells (Figure 4a). This was accompanied by a rise in lactate/pyruvate ratios (Figure 4b, *p* < 0.0001), a decline in malate/oxaloacetate ratios (Figure 4c, *p* < 0.001) and a reduction in mitochondrial oxygen consumption rates (OCRs, Appendix A, *p* < 0.05) in asparagine-starved A-N0 compared to control C-N5 and WT W-N5 cells. Increased pyruvate-to-lactate and malate-to-oxaloacetate conversion by lactate dehydrogenase and malate dehydrogenase, respectively, may contribute to the regeneration of NAD^+^ and alleviate the relative lack of electron acceptors. Again, supplementation with excess asparagine partially reverted NAD^+^/NADH, lactate/pyruvate and malate/oxaloacetate ratios to levels similar to those in control C-N5 cells (Figure 4a–c).

### 2.6. Supplementation with Pyruvate Reverses Reductive Stress and Global Metabolomic Changes in Asparagine-Starved Sarcoma Cells

Findings from our experiments indicate that asparagine starvation of sarcoma cells causes a relative lack of electron acceptors through relative accumulation of NADH (reductive stress). Previously published studies demonstrated that exogenous electron acceptors such as pyruvate rescued the growth arrest associated with reductive stress in tumor cells [13,14,15]. Therefore, asparagine-deprived shASNS cells (A-N0) were grown in medium containing 0 mg/L asparagine, 5 mg/L asparagine and 100 mg/L asparagine and supplemented with exogenous pyruvate at a concentration of 88.06 mg/L (1 mM) (Figure 4d).

Indeed, pyruvate supplementation resulted in a rise in NAD^+^/NADH ratios in A-N0 cells (*p* < 0.001) (Figure 4a), whereas exogenous pyruvate did not change the NAD^+^/NADH ratio in shASNS cells supplemented with excess amounts of asparagine (A-N100, ns), in control shLuc (C-N5, ns) or in wild-type cells (W-N5, ns) (Figure 4a). Increased regeneration of NAD^+^ in asparagine-deprived cells (A-N0) after supplementation with exogenous pyruvate was accompanied by a decline in lactate/pyruvate (Figure 4b, *p* < 0.001) and a rise in malate/oxaloacetate ratios (*p* < 0.05, Figure 4c).

Furthermore, the global metabolome of asparagine-starved A-N0 cells was evaluated and compared to control cells with and without exogenously supplemented pyruvate by GC-MS (Figure 4d–f). Principal component analyses again showed marked differences in the metabolomic profile of asparagine-starved shASNS cells (A-N0; marked in red) compared to control shLuc/WT cells (C-N5, W-N5; marked in shades of blue). Notably, asparagine-starved shASNS cells supplemented with exogenous pyruvate (A-N0-P; marked in pink) clustered with the metabolome of control shLuc/wild-type cells (C-N5, W-N5; marked in shades of blue). Exogenous pyruvate did not change the metabolomic profile of control shLuc/wild-type cells (C-N5 and W-N5 versus C-N5-P and W-N5-P; marked in shades of blue) and shASNS cells supplemented with excess asparagine (A-N100 versus A-N100-P; marked in shades of green) (Figure 4e, Appendix A). Thus, pyruvate supplementation rescued the global metabolomic changes induced by asparagine deprivation (Figure 4e). This included changes in TCA cycle metabolites: isocitrate, succinate, citrate, malate, aconitate, oxaloacetate, fumarate and alpha-ketoglutarate levels were reduced in asparagine-starved cells (A-N0, marked in red) versus control shLuc cells (C-N5, marked in blue (Figure 4f). Yet, isocitrate, succinate, citrate, malate, aconitate, oxaloacetate, fumarate and alpha-ketoglutarate levels in asparagine-deprived cells supplemented with pyruvate (A-N0-P, marked in pink) were restored to levels comparable to those in control cells (Figure 4f). For example, the glutamine content in asparagine-deprived cells supplemented with pyruvate (A-N0-P) was restored to levels similar to those in control cells (C-N5) (Figure 4g). Aspartate levels in asparagine-depleted cells were also raised by exogenous addition of the electron acceptor pyruvate, but the increase did not reach statistical significance (Figure 4h). Aspartate/glutamine ratios in asparagine-starved cells supplemented with pyruvate (A-N0-P) were restored to levels comparable to those in control cells (C-N5) (Figure 4i).

### 2.7. Supplementation with Pyruvate Reverses the Growth-Inhibitory Effects of Asparagine Depletion in Mouse Sarcoma Cells

Asparagine depletion of sarcoma cells can also be achieved by exposure to asparaginase (N-ase), an U.S. Food and Drug Administration (FDA)-approved drug which catalyzes the breakdown of asparagine and glutamine in medium with and without cells (Figure 5a) [16]. Purwaha et al. previously demonstrated that 0.1 and 0.5 U/mL asparaginase had similar effects on the asparagine and glutamine content in medium after 48 h [17]. Also, asparaginase was previously shown to reduce the growth of mouse and human sarcoma cells (Figure 5h–i) [3]. Asparaginase’s effects on the global metabolomic profile of mouse sarcoma cells grown with and without exogenously supplemented asparagine (100 mg/L) were investigated by GC-MS (Figure 5b). Asparaginase-treated sarcoma cells supplemented with exogenous asparagine (W-A-N100, marked in green) clustered with asparaginase-treated cells cultured in asparagine-free medium (W-A-N0, marked in red), but there were marked differences between asparaginase-treated (W-A-N0, W-A-N100) and control sarcoma cells (W-0-N0, W-0-N100, marked in shades of blue) (Figure 5c, Appendix A).

Differentially regulated metabolites included lower citrate (Figure 5d,g), alpha-ketoglutarate (Figure 5e,g) and malate (Figure 5f–g) levels in asparaginase-treated cells (W-A-N0, W-A-N100) compared to control cells (W-0-N0, WT-0-N100). These differences were accompanied by lower aspartate, glutamate and glutamine content in asparaginase-treated cells (W-A-N0, W-A-N100) compared to control cells (W-0-N0) (Figure 5g) and reminiscent of the changes observed in shASNS sarcoma cells cultured in asparagine-free medium (Figure 3 and Figure 4). Notably, exogenous supplementation with pyruvate, which was previously shown to reverse reductive stress and global metabolomic changes in asparagine-starved sarcoma cells (Figure 4), fully reversed the growth-inhibitory effects of 0.3 U/mL asparaginase on mouse sarcoma cells (Figure 5h). Exogenous supplementation with aspartate partially reversed the growth-inhibitory effects of asparaginase on mouse sarcoma cells (Figure 5i).

### 2.8. Complex 1 Inhibitors Augment the Anti-Proliferative Effects of Asparagine Depletion on Mouse Sarcoma Cells

The mitochondrial electron transport chain (ETC) consists of four enzyme complexes that transfer electrons from donors such as NADH to oxygen (Figure 6a). Chemical disruption of the ETC by complex 1 inhibitors has long been known to impede regeneration of electron acceptors and block cell proliferation [13,14,15]. We exposed mouse sarcoma cells to low concentrations of phenformin (10 µM), which did not alter proliferation activity (Figure 6b, ns). Yet, when mouse sarcoma cells were treated concomitantly with 10 µM phenformin and 0.3 U/mL asparaginase, phenformin significantly augmented the anti-proliferative effects of asparaginase alone (Figure 6b, *p* < 0.0001). Two alternative complex 1 inhibitors (metformin and imiquimod at 1 mM and 20 µM, respectively) also reduced mouse sarcoma cell proliferation and deepened the anti-proliferative effects of asparaginase (Figure 6c–d *p* < 0.0001). The growth-inhibitory effects of asparaginase, all three complex 1 inhibitors and combinatorial treatment with asparaginase and complex 1 inhibitors were reversed by supplementing cells with the exogenous electron acceptor pyruvate (1 mM, Figure 6b–d). We conclude that chemical disruption of the ETC augments the anti-proliferative effects of asparaginase exposure, likely by deepening the relative excess of NADH/reducing equivalents caused by asparagine deprivation.

### 2.9. Complex 1 Inhibitors Augment Reductive Stress and Growth Inhibition Induced by Asparagine Deprivation of Human Sarcoma Cells

Analogous to what was observed in mouse sarcoma cells, asparagine depletion was shown to reduce the growth of human sarcoma cell lines [3], including the growth of the *NRAS*-mutated rhabdomyosarcoma cell line RD [18]. The anti-proliferative effects of asparagine deprivation were confirmed in shASNS RD cells (A-N0) cultured in asparagine-free conditions compared to shASNS RD cells (A-N100, *p* < 0.0001) in supraphysiological asparagine conditions and shLuc control cells in media with and without supplemental asparagine (C-N100, C-N0, *p* < 0.0001, Figure 7a,b). The experiment was performed in physiological (73.1 mg/L, Figure 7a) and supraphysiological (584.6 mg/L, Figure 7b) glutamine environments. Of note, the anti-proliferative effects of asparagine depletion were similar (*p* > 0.00001) and partially reversed by supplementation with pyruvate (*p* < 0.05) in physiological and supraphysiological glutamine conditions (Figure 7a,b). However, supplementation with asparagine partially restored proliferation of asparagine-starved human sarcoma cells in the physiological glutamine environment only (Figure 7b, *p* > 0.0001), not in the supraphysiological glutamine conditions (Figure 7a, ns).

We demonstrated that asparaginase, which hydrolyzes both asparagine and glutamine in the cell environment, reduced the growth of human RD cells at concentrations of 1 and 0.3 U/mL (Figure 7c, *p* < 0.0001), while supplementation with exogenous 1 mM pyruvate reversed the anti-proliferative effects of asparaginase treatment (Figure 7c, *p* < 0.0001). Asparaginase also raised the percentage of Annexin V-positive/7AAD-negative apoptotic RD cells (Figure 7d–e, *p* < 0.01). As complex 1 inhibitors significantly augmented the adverse effects of asparagine depletion of mouse sarcoma cells (Figure 6), human RD cells were exposed to combinatorial treatment with asparaginase and phenformin (Figure 7d–f).

Combinatorial treatment with asparaginase and phenformin further augmented the pro-apoptotic effects of either drug alone (Figure 7e, *p* < 0.05). Similarly, combinatorial treatment with asparaginase and phenformin deepened the anti-proliferative effects of asparaginase or phenformin alone (Figure 7f, *p* < 0.0001). Analogous to our observations in mouse sarcoma cells treated with asparaginase and complex 1 inhibitors (Figure 6), supplementation with the electron acceptor pyruvate reversed the anti-proliferative effects of asparaginase and phenformin alone and in combination on human RD cells (Figure 7f, *p* < 0.01). Analogous experiments confirmed that metformin or imiquimod also augmented the anti-proliferative (Appendix A, *p* < 0.01) and pro-apoptotic (Appendix A, *p* < 0.01) effects of asparaginase on human RD sarcoma cells. Again, supplementation with pyruvate reversed the anti-proliferative effects of asparaginase and metformin (Appendix A)/imiquimod (Appendix A) alone and in combination on human RD cells. Of note, we also used four distinct primary human rhabdomyosarcoma cell cultures (Appendix A) to confirm that simultaneous exposure to asparaginase and imiquimod significantly enhanced the anti-proliferative effects of either asparaginase or imiquimod alone in patient-derived low-passage sarcoma cells (Figure 8a–d).

Finally, LC-MS was used to determine changes in the NAD^+^/NADH ratio in human RD cells exposed to phenformin and asparaginase. NAD^+^/NADH ratios were reduced in RD cells treated with asparaginase alone (*p* < 0.01), phenformin alone (*p* < 0.0001) and asparaginase and phenformin in combination (*p* < 0.0001) compared to control cells (Figure 7g). In fact, combinatorial treatment with asparaginase and phenformin caused a higher reduction in NAD^+^/NADH ratios than that induced by asparaginase alone (Figure 7g, *p* < 0.0001). Supplementation with pyruvate partially restored NAD^+^/NADH ratios in RD cells treated with phenformin alone (*p* < 0.01). Pyruvate also raised the NAD^+^/NADH ratios in RD cells treated with asparaginase alone or asparaginase and phenformin in combination, but these changes did not reach statistical significance (Figure 7g).

## 3. Discussion

Rapidly proliferating cancer cells have to meet specific metabolic requirements to sustain homeostasis while supporting rapid expansion of biomass, which poses a profound metabolic challenge [19]. In this study, we employed different mass spectrometry approaches to identify lower aspartate levels, higher aspartate/glutamine ratios and lower levels of TCA cycle metabolites in asparagine-deprived cells. These changes indicated a redirection of TCA cycle flux and were accompanied by reduced NAD^+^/NADH ratios, consistent with a relative lack of electron acceptors. Reductive stress caused by deficiency of electron acceptors may be just as harmful as oxidative stress, as electron acceptors NAD^+^, NADP^+^ and GSH are essential for maintaining cellular homeostasis. For example, NAD^+^ is a necessary cofactor for many enzymes, and a decrease in the NAD^+^/NADH ratio causes these enzymes to decrease in activity [20]. We discovered that asparagine depletion of sarcoma cells causes reductive stress and that exogenous supplementation with the electron acceptor pyruvate [13,14,15] restored the changes in NAD^+^/NADH ratios, proliferation and viability induced by asparagine deprivation. Treatment with chemicals disrupting the regeneration of NAD^+^ in the ETC further enhanced the anti-proliferative and pro-apoptotic effects of asparagine depletion.

Aspartate is known to serve as an important carbon donor for the synthesis of proteinogenic amino acids and nucleotides. Our metabolomic analyses demonstrated consistently reduced aspartate levels in asparagine-deprived cells, including shASNS cells grown in asparagine-free medium and asparaginase-treated cells. This was surprising at first as we had initially expected higher aspartate levels due to reduced aspartate consumption after ASNS knockdown or increased release of aspartate by asparaginase. Yet, addition of the electron acceptor pyruvate not only reversed reductive stress and growth arrest in asparagine-starved cells but also raised aspartate levels. Thus, we conclude that asparagine deprivation, through metabolic reprogramming, causes reductive stress, which, in turn, results in lower aspartate levels in asparagine-starved cells.

Growth of various types of tumor cells was recently shown to depend on adequate availability of the non-essential amino acid asparagine [3,5,10]. Mammalian cells maintain intracellular asparagine levels through ATP-dependent conversion, catalyzed by the enzyme ASNS, from aspartate and glutamine into asparagine and glutamate (Figure 1) [6]. The growth-inhibitory effects of ASNS silencing are reversible through uptake of exogenously supplemented asparagine [3]. Findings from our experiments confirm that asparagine deprivation stalls proliferation, induces autophagy and increases apoptosis in sarcoma cells grown in physiological and supraphysiological glucose and glutamine conditions. Consistent with the published literature [10], asparagine did not reverse the growth inhibition observed in glutamine-starved sarcoma cells. Yet, exogenous asparagine partially reversed glutamine withdrawal-induced apoptosis in *KRAS*-driven mouse sarcoma cells. Zhang et al. also reported complete reversal of glutamine withdrawal-induced apoptosis by supplemental asparagine in glioma and neuroblastoma cells [10].

Thus, asparagine appears to play an important role in cancer cell metabolism [3,5,10]. ASNS transcription in cancer cells is controlled by tumor-driving oncogenes [4,21] and elevated in response to amino acid and glucose starvation through adaptive processes known as amino acid and endoplasmic reticulum stress responses [6]. These cellular programs converge on increased translation of the transcription factor ATF4, which, in turn, stimulates expression of ASNS and various other enzymes and nutrient transporters [4,6]. We speculate that solid tumors express high levels of ASNS to maintain intracellular asparagine supply as they outgrow nutrient supply via the existing vasculature and thereby promote tumor cell survival. This is further supported by correlative studies indicating that higher ASNS expression in tumor cells correlates with higher tumor grade, higher rates of metastases and worse patient outcomes [9,10]. However, asparagine does not appear to serve as a major intermediary metabolite. Flux analyses demonstrated that 10% of its nitrogen and minimal amounts of its carbon were detected in purines and aspartate/malate, respectively [5]. Nevertheless, asparagine is an important proteinogenic amino acid [3,10], especially with respect to the production of asparagine-rich proteins involved in epithelial-to-mesenchymal transition [9]. We demonstrate that transcripts involved in nascent peptide synthesis (e.g., KEGG categories ribosome and aminoacyl tRNA biosynthesis) are enriched among genes differentially regulated in asparagine-starved cells. Furthermore, intracellular asparagine serves as an amino acid exchange factor to secure intracellular levels of other amino acids, especially serine/threonine and non-polar amino acids [5]. In fact, asparagine appears to coordinate protein and nucleotide synthesis in proliferating cells by regulating mTORC1 activity [4,21] and uptake of serine [4], which is crucial for protein and pyrimidine synthesis.

Asparagine metabolism in cancer cells represents a vulnerability with possible therapeutic value. Asparagine-depleting drugs have been long established as effective drugs in the treatment of lymphoblastic leukemia [6]. Asparaginase was also shown to reduce sarcoma growth. However, when compared to lymphoblasts, asparaginase sensitivity of sarcoma cells is moderate to poor [3,11]. This may be due to the fact that sarcoma cells and many other solid tumor cells express substantially higher levels of ASNS than lymphoblasts [9,12], which are characterized by low ASNS levels and are much more dependent on environmental asparagine. Still, asparaginase is an attractive anti-cancer drug, as its side effect profile does not overlap with the toxicities of many other conventional chemotherapy drugs. It may be a valuable component in anti-sarcoma combination therapies. Synergistic/additive effects were previously reported for asparaginase and mTOR inhibitors [21], autophagy inhibitors such as chloroquine [22] and proteasome inhibitors such as bortezomib [23]. Chemical inhibitors of the mitochondrial electron transport chain (ETC) were previously shown to decrease NAD^+^/NADH ratios and halt growth in tumor cells [13,14]. This led us to examine if complex 1 inhibitors deepened the reductive stress and growth arrest caused by asparagine depletion. Our experiments revealed a clear synergistic effect of asparaginase and complex 1 inhibitors on the growth of mouse and human sarcomas, including patient-derived rhabdomyosarcoma cultures. Complex 1 inhibitors may be applied in combination with asparaginase to further exploit the anti-cancer effects of asparagine depletion in sarcomas.

## 4. Materials and Methods

### 4.1. Cell Lines

The *KRAS*-driven mouse sarcoma cell line used in this study was established from a genetically engineered Kras(G12v);p16p19^null^ mouse sarcoma with myogenic differentiation [3,12]. The human embryonal rhabdomyosarcoma cell line RD was purchased from American Type Culture Collection (ATCC). Both cell lines were kept in DMEM with 10% fetal bovine serum (FBS, Sigma Aldrich, St. Louis, MO, USA) and 1% penicillin–streptomycin (PS). To interrogate asparagine effects, 4.15 g DMEM (D5030, Sigma Aldrich), 1.85 g NaHCO3 (Sigma Aldrich), 10% FBS and 1% PS were reconstituted in 500 mL dH2O together with defined amounts of glutamine (0 to 292 mg; Sigma), glucose (0.25 to 2.25 g; Sigma Aldrich) and asparagine (0 to 50 mg; A4159, Sigma Aldrich). Short tandem repeat analyses of mouse (Appendix A) and human (Appendix A) cell lines used in this study were performed by Eurofins.

RMSZH003_XC primary rhabdomyosarcoma cell cultures were established from a recurrent pelvic PAX3:FOXO1 fusion-positive rhabdomyosarcoma diagnosed in a 3-year-old female, IC-pPDX-29_XC from a 14-year-cold female with a relapsed, paravertebral PAX3:FOXO1 fusion-positive rhabdomyosarcoma, SJRHB011_YC from a 4-year-old boy with a head/neck fusion-negative rhabdomyosarcoma and SJRHB012_ZC from an 18-year-old male with a relapsed bladder/prostate fusion-negative rhabdomyosarcoma (Appendix A). Primary rhabdomyosarcoma cell cultures were maintained in neurobasal medium (Gibco, 21103049, Waltham, MA, USA) supplemented with 1% penicillin/streptomycin (Gibco, 15140-122), 1X Glutamax (Gibco, 35050) and 2X B27 (Life Technologies, 17504044, Carlsbad, CA, USA). For RMS-ZH003_XC and IC-pPDX-29_XC cells, 20 ng/ml basic fibroblast growth factor (bFGF) (Peprotech, AF-100-18B, Rocky Hill, NJ, USA) and 20 ng/mL epidermal growth factor (EGF) (Peprotech, AF-100-15) were added [24].

### 4.2. ASNS Silencing

TRC clones TRCN0000324779, TRCN0000031703 and TRCN0000031702, delivered in PLKO vectors, were employed to knock down ASNS in mouse sarcoma cells as previously described [3].

### 4.3. Proliferation Assays

Cells were seeded at 750 (mouse) or 1000 (human RD) cells per well in 96-well plates with 100 µL medium. After 24 h, cells were exposed to asparaginase (MBS142428, Biozol), phenformin (14997, Cayman Chemical, Ann Arbor, MI, USA), metformin (13118, Cayman Chemical) and imiquimod (14956, Cayman Chemical) at defined concentrations. Chemical treatments were repeated every 48 h for up to 6 days in total. To determine cell growth, 3-(4,5-dimethylthiazol-2-yl)-2,5-diphenyltetrazolium bromide (MTT) (M2003, Sigma-Aldrich) was solved in PBS to a final concentration of 12 mM. In brief, 10 µL of MTT solution was added per well and incubated for 4 h. Then, 100 µL dissolving solution (10% SDS (L3771, Sigma-Aldrich) with 0.01M HCL) was added to each well and incubated for 12–16 h. Optical densities at 570 nm were measured with a Tecan Sunrise absorbance microplate reader.

### 4.4. Cell Viability Assay

SJRHB012_ZC, SJRB011_YC, IC-pPDX-29_XC and RMS-ZH003_XC cells were plated at 6000 cells/well in 384-well plates coated with Matrigel. The medium was changed after one day, and cells were incubated with either asparaginase 1 u/mL (MBS142428, Biozol, Eching, Germany), imiquimod 50 uM (14956, Cayman Chemical) or both. DMSO-treated cells served as controls. After 72 h, cell viability was determined using the CellTiter-Glo 3D viability assay (Promega, G9681, Madison, WI, USA).

### 4.5. Annexin V Staining

Annexin V staining was performed according to the manufacturer’s instructions using Annexin V-APC (550474, BD Biosciences, San Jose, CA, USA) and 7AAD (559925, BD Biosciences) with a BD FACS Canto II flow cytometer.

### 4.6. Western Blots

Cells were detached using trypsin, washed twice in ice-cold PBS, lysed for 15 min on ice using lysis buffer (9803S, NEB, Ipswich, MA, USA) containing protease and phosphatase inhibitors (5872 S, Cell Signaling Tech, Denver, MA, USA) and centrifuged for 10 min at 13,000 *g*. Protein concentrations in supernatants were measured using the DC Protein Assay (Bio-Rad, # 5000111). Equal amounts of protein extracts were processed for Western blots. Specific antibodies and concentrations were used to detect expression of asparagine synthetase (1:250, Sigma Aldrich, HPA029385), LC3I/II (1:500, Cell Signalling Technology, #1241) and actin (1:10,000, Sigma Aldrich, A5441). Immune complexes were detected by chemiluminescence (ECL, RPN2235, GE Healthcare Life Sciences, Marlborough, MA, USA).

### 4.7. RNA Isolation, RNA-Seq and qRT-PCR

Total RNA was isolated using TRIzol Reagent (Ambion, 15596018, Waltham, MA, USA) and quantified using a 2.0 Qubit fluorometer (Invitrogen, Carlsbad, CA, USA). RNA integrity was confirmed using an Agilent 2100 Bioanalyzer.

RNA-Seq was performed in duplicate. Paired-end reads were aligned to the mouse genome (mm10) and quantified after adapter removal and bad quality trimming with Trimmomatic [25] and STAR aligner [25,26]. Identification of the differentially regulated genes between A-N0 and A-N5 cells versus A-N100 and C-N5 cells was done using the linear-based model limma voom [27]. Gene set enrichment analysis was performed with GAGE [28] using gene sets from MSigDB [29]. Human Entrez IDs were converted to their mouse ortholog Entrez IDs. For all analyses, the significance thresholds were set to an adjusted *p*-value of 0.05. Raw data are accessible on GEO (GSE153991; token: ozgdacsyhihnkf).

cDNA synthesis was performed using the Superscript III First Strand kit from Invitrogen. Reverse transcription was achieved using the Superscript III First-Strand Synthesis System, and qRT-PCR was conducted using an ABI7900 RT-PCR system (Applied Biosystem) with SYBR Green PCR reagents. Primer sequences are listed in Appendix A.

### 4.8. Metabolite Extraction

Metabolite extraction and quenching were performed on plate with 1.5 mL ice cold extraction medium (90% methanol, 10% water) containing 1 µg/mL ribitol (A5502, Sigma Aldrich), phenyl β-d-glucopyranoside (292710, Sigma Aldrich), isoguanosine (sc-207768, Santa Cruz, Dallas, TX, USA), d4-succinate (293075, Sigma Aldrich) and methyl-tyrosine (M8131, Sigma Aldrich) as the internal standard. Cells were detached on ice by using a cell scraper, transferred into screw-cap tubes prefilled with 300 mg glass beads (G4649, Sigma Aldrich) and immediately frozen in liquid nitrogen until homogenization. Cells were homogenized using a Precellys tissue homogenizer (P000669-PR240-A, Bertin instruments, Montigny-le-Bretonneux, France) at −10 °C. Three cycles of homogenization for 15 s at 6500 rpm were applied with 10-s breaks in between cycles. Samples were then centrifuged (20,000× *g*) for 10 min at 4 °C to remove cell debris and protein precipitates, and 500 µL of each supernatant was transferred into two new reaction tubes for GC-MS and LC-MS analyses. Finally, extracts were dried using a vacuum rotator (Eppendorf, Hamburg, Germany) and flushed with nitrogen. The DNA content in the extracts was measured using NanoDrop 1000 (Thermo Fisher Scientific).

### 4.9. Gas Chromatography–Mass Spectrometry (GC-MS)

To achieve methoxymethylation of keto- and aldehyde groups and trimethylsilylation of amines, hydroxyl groups and carboxylic groups, dried metabolite pellets were derivatized by adding 20 µL methoxyamine hydrochloride (226904, Sigma Aldrich) (20 mg/mL in pyridine) and incubated for 90 min at 28 °C at 1200 rpm and then incubated with 50 µL *N*-methyl-*N*-(trimethylsilyl)-trifluoroacetamide (69479, Merck, Kenilworth, NJ, USA) for 30 min at 37 °C and 1200 rpm. GC-MS analysis was performed using an Agilent 7890 A/5975 C system with a Gerstel MPS2 XL autosampler. An HP5-MS column (5% diphenyl–95% dimethylpolysiloxane, 60 m × 0.25 mm × 0.25 μm) (AG19091S-436E, Agilent, Santa Clara, CA, USA) was used for GC separation (80 °C for 3 min, increase to 320 °C at 5 °C per minute, then hold for 14 min with a carrier gas flow rate of 1 mL/minute and septum purge flow rate of 3 mL/minute). Full scan spectra were acquired from 50 to 800 m/z at a scan rate of 1.99 per second. Equilibration time and post-run time were set to 1 min, inlet temperature to 230 °C, MS source temperature to 230 °C and quadrupole analyzer temperature to 150 °C. Pooled samples were injected at regular intervals to serve as internal controls. Raw data are accessible on https://www.ebi.ac.uk/metabolights/MTBLS2035 [30].

### 4.10. GC-MS Data Analysis

A system-independent Kovats retention index (RI) was generated by using one C10-C40 n-alkane standard sample within the GC-MS sample set. Peak identification, deconvolution and integration were performed by the automated mass spectral deconvolution and identification system (AMDIS) version 2.72 with the following parameters: component width: 12; omit TIC; adjacent peak subtraction: one; resolution: medium; sensitivity: medium; shape requirements: medium. Each sample was processed as a batch job and ELU files were saved as AMDIS outputs and further processed by the SpectConnect online service to generate a compound matrix [31]. SpectConnect settings were set as follows: elution threshold, 1 min; support threshold, 50%; similarity threshold, 80%. Compound annotation was performed by matching the obtained spectra with the library spectra of FiehnLib [32], Nist14 [33], Golm DB [34] and an in-house database. Compounds were accepted with a match factor threshold of more than 750 and a retention index deviation of less than 5%. Normalization was performed using the total peak area of the chromatogram with phenyl β-d-glucopyranoside or ribitol as the internal standard. Division of the total peak areas of the chromatogram was previously shown to be an approximation for the total cell number [35].

### 4.11. Liquid Chromatography–Mass Spectrometry (LC-MS)

LC-MS was used to evaluate the amino acid spectrum, NAD^+^/NADH ratios and TCA cycle metabolites.

For targeted amino acid analyses, cell extracts were directly injected onto a BEH Amide 1.8 µm, 150 × 2.1 mm column using two buffers: 0.1% formic acid (4724.1, Sigma Aldrich) was added to water (buffer A) and acetonitrile (HN40.2, Carl Roth, Karlsruhe, Germany; buffer B). Separation took place using the following program: 90% B to 85% B in 0.2 min, 85% B to 75% B in 0.8 min, 75% B to 40% B in 1 min, 40% B to 50% B in 0.1 min, 50% B for 5.9 min, 50% B to 90% B in 0.2 min and 90% B for 4.8 min. The flow rate was set to 600 µL per minute and column temperature to 50 °C. Methyl-tyrosine served as an internal standard. Pooled samples were injected at regular intervals as internal controls.

For NAD^+^/NADH-specific analyses, cell extracts were injected onto a Waters acquity HSS T3 1.8 µm, 100 × 2.1 mm column using two buffers: 0.1% formic acid was added to water (buffer A) and methanol (buffer B). Separation took place using the following program: 100% A for 1.5 min, 100% A to 95% A in 2.5 min, 95% A to 5% A in 4 min, 5% A for 8 min, 5% A to 100% A in 0.2 min and 100% A for 7.8 min. The flow rate was set to 200 µL per minute and the column temperature was set to 50 °C. Pooled samples were injected at regular intervals as internal controls.

TCA cycle metabolites were analyzed according to Tan et al. [36]. Pooled samples were injected at regular intervals as internal controls. Raw data are accessible on https://www.ebi.ac.uk/metabolights/MTBLS2035 [30].

### 4.12. LC-MS Data Analysis

LC-MS data analysis was performed using Quantitative Analysis software (Agilent) with specific MRM transitions, qualifier ratios and retention times for metabolite identification. Data were normalized using internal standards and the total peak area of the corresponding sample was obtained by GC-MS or DNA concentration. Division of total peak areas of a chromatogram was previously shown to be an approximation for the total cell number [35].

### 4.13. Oxygen Consumption Rate

Oxygen consumption rate (OCR) was measured using a Seahorse XF96 analyzer (Agilent). Cells were seeded 48 h before the experiment. Cells were exposed to phenol red- and bicarbonate-free DMEM with 292.3 mg/L glutamine and 1.80 g/L glucose immediately prior to the experiment. Cells were incubated for at least one hour at 37 °C in a non-CO_2_ incubator. The mito stress test assay was performed at final concentrations of oligomycin at 3 µM, FCCP at 1 µM, antimycin A at 2 µM and rotenone at 2.5 µM.

### 4.14. Statistics

MS data were analyzed using MetaboAnalyst 3.0 [37,38]. Missing values were replaced by the half minimum positive value. All experiments were replicated two to three times.

For the GC-MS analyses on asparagine-deprived compared to control mouse sarcoma cells presented in Figure 3, samples were run in three independent experiments with 4 replicates per condition. Each of the 3 experiments included wild-type control cells grown under physiological asparagine concentrations (W-5). For each experiment, MS data from W-5 runs were averaged and then used to normalize MS data obtained for the other samples. If ≥2 out of 4 W-5 replicates did not detect specific metabolites, these were excluded from the analysis. For all other conditions, missing values were replaced by the half minimum positive value. Only metabolites which were detected in all three experiments were included in the overall analysis.

For the GC-MS analyses on asparagine-deprived mouse sarcoma cells cultured with and without supplemental pyruvate, for the GC-MS analyses on asparaginase- and vehicle-treated mouse sarcoma cells and for the LC-MS analyses, samples were run as part of one single experiment each. Range scaling (mean-centered and divided by the range of each variable) was used for principal component analysis and heat map generation. Cluster and distance analyses were conducted following Pearson and Ward. Differentially regulated metabolites were determined using a one-way analysis of variance (ANOVA) with an adjusted *p*-value (FDR) cutoff of 0.05. Differences in mean values were tested for statistical significance using Tukey’s HSD post-hoc test.

Differences in cell growth, apoptosis and transcript expression were tested for statistical significance using t-tests or ANOVAs with an adjusted *p*-value (FDR) cut-off of 0.05. Differences in mean values were determined using Tukey’s HSD post-hoc tests.

## 5. Conclusions

Asparagine does not serve as an intermediary metabolite, but it still assumes an important role in sarcoma metabolism. Adequate asparagine availability is required to support sarcoma growth. Our metabolomic studies discovered reductive stress resulting in consistently low aspartate levels in asparagine-deprived sarcoma cells. We conclude that reductive stress appears to be a major contributor to the growth arrest observed in asparagine-starved cells. Further studies are needed to elucidate the cellular processes that contribute to metabolic reprogramming, ultimately resulting in a lack of electron acceptors in asparagine-starved tumor cells.

## Figures and Tables

**Figure 1 cancers-13-00412-f001:**
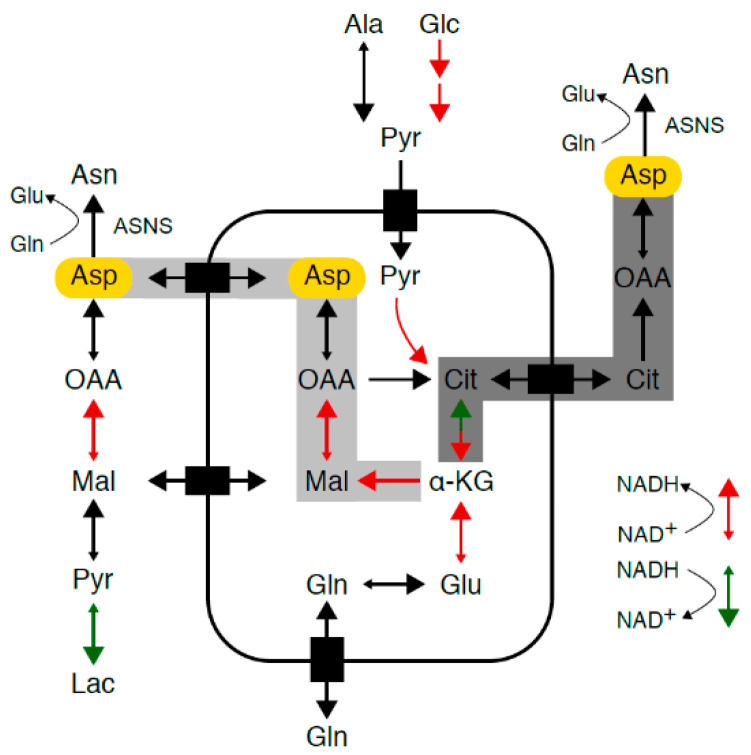
Asparagine synthesis. Mammalian cells maintain intracellular asparagine availability through import from the extracellular space and de novo synthesis from glutamine and aspartate (Asp, marked in yellow) through a unidirectional ATP-dependent reaction catalyzed by asparagine synthetase (ASNS). As aspartate transport into most mammalian cells is inefficient, aspartate supply depends on conversion of glutamine to glutamate to alpha-ketoglutarate, which enters the tricarboxylic acid (TCA) cycle and is converted into aspartate via reductive carboxylation (marked in dark gray) or oxidative decarboxylation (marked in light gray). TCA cycle flux and glycolysis are accompanied by a continuous flow of electrons, during which nicotinamide adenine dinucleotide (NAD^+^) serves as an electron carrier in a continuous cycle of reduction to nicotinamide adenine dinucleotide hydride (NADH) (marked in red) and oxidation back to NAD^+^ (marked in green).

**Figure 2 cancers-13-00412-f002:**
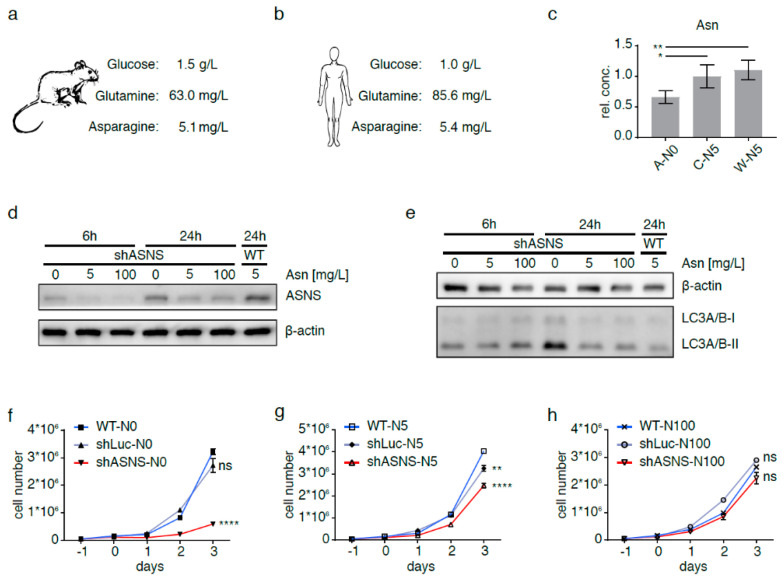
Asparagine depletion in mouse sarcoma cells. Physiological glucose, glutamine and asparagine concentrations in (**a**) mouse and (**b**) human plasma. Intracellular asparagine was depleted in mouse sarcoma cells by abrogating asparagine synthesis via shRNA-mediated ASNS knockdown and culture in asparagine-free medium (A-N0). (**c**) Reduction in intracellular asparagine content in A-N0 cells to <66% of that in shLuc (C-N5) and wild-type (W-N5) sarcoma cells grown in medium containing near-physiological (5 mg/L) asparagine. (**d**) Minimal ASNS expression in shASNS cells cultured in medium with 5 mg/L or excess (100 mg/L) asparagine; upregulation of ASNS expression in shASNS cells grown in asparagine-free (0 mg/L) medium. Raw data please see Appendix A. (**e**) Increased processing of LC3I/II in shASNS cells grown in asparagine-free medium for 6 and 24 h. Raw data please see Appendix A. (**f**–**g**) The growth of shASNS sarcoma cells (red line) was significantly reduced compared to the growth of wild-type cells (blue line) and shLuc control cells (gray line) (**f**) in asparagine-free medium and (**g**) in medium containing 5 mg/L asparagine. (**h**) In medium containing excess asparagine, shASNS and wild-type sarcoma cells grew equally well. Please see Appendix A for asparagine content of cells measured by LC-MS. Please see Appendix A (Raw data in Appendix A) and Appendix A for the impact of asparagine starvation on proliferation, apoptosis and autophagy of mouse sarcoma cells in the context of high, physiological and low glutamine (584.6, 73.1 and 7.3 mg/L, respectively) and high and low glucose (4.5 and 0.5 g/L, respectively) concentrations. Data were evaluated for statistical significance by one-way ANOVA with Tukey’s post-hoc test (ns *p* ≥ 0.05, * *p* < 0.05, ** *p* < 0.01, **** *p* < 0.0001).

**Figure 3 cancers-13-00412-f003:**
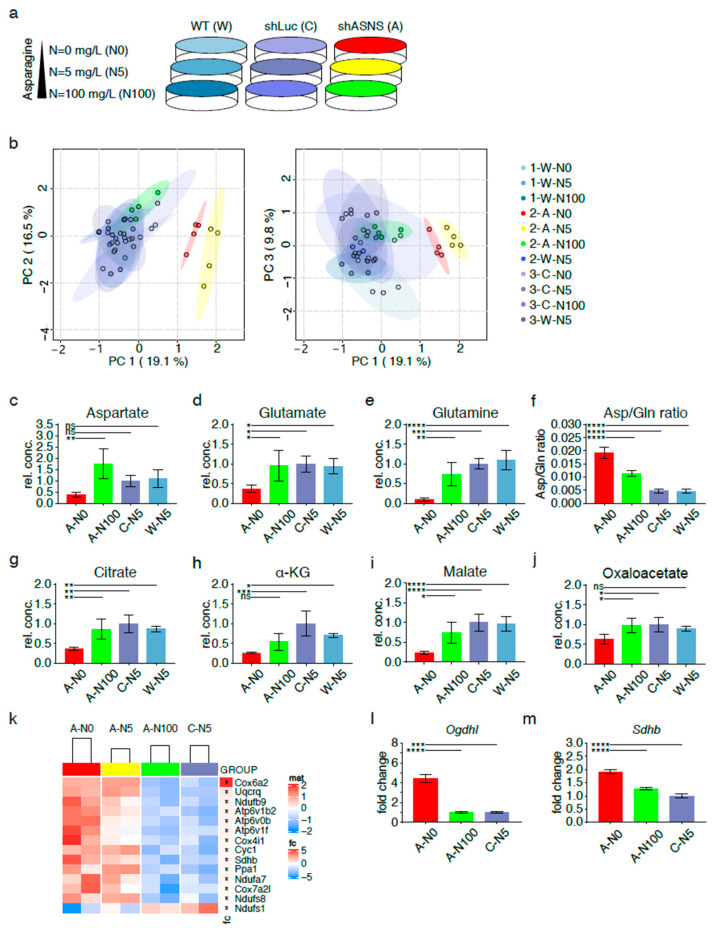
Metabolic adaptation of asparagine-deprived mouse sarcoma cells. (**a**) Asparagine availability was modified by exposing shASNS (A) sarcoma cells, shLuc control (C) sarcoma cells and wild-type (W) sarcoma cells to asparagine-free medium, medium containing near-physiological (5 mg/L) or supraphysiological asparagine concentrations (100 mg/L) for 2 days. Global metabolomic changes were investigated by GC-MS: samples were run in three independent experiments with four replicates per condition (experiment 1: 1-W-N0, 1-W-N5, 1-W-N100; experiment 2: 2-A-N0, 2-A-N5, 2-A-N100, 2-W-N5; experiment 3: 3-C-N0. 3-C-N5. 3-C-N100, 3-W-N5). Each of the three experiments included wild-type control cells grown at physiological asparagine concentrations (1-W-5, 2-W-N5, 3-W-N5). For each experiment, MS data from W-5 runs were averaged and then used to normalize MS data obtained for the other samples. (**b**) Principal component analyses highlighted global differences in the metabolome of shASNS cells cultured with 0 (2-A-N0; marked in red) or 5 mg/L (2-A-N5; marked in yellow) asparagine as opposed to shLuc and wild-type control cells (marked in shades of blue). After supplementation with 100 mg/L asparagine (2-A-N100, marked in green), shASNS cells clustered with shLuc and wild-type cells. Further analyses by LC-MS revealed that the metabolome of asparagine-deprived A-N0 sarcoma cells grown without supplemental asparagine was marked by (**c**) low aspartate, (**d**) glutamate and (**e**) glutamine levels compared to control C-N5 and W-N5 cells. (**f**) Higher aspartate/glutamine ratios suggest that glutamine is shunted towards aspartate synthesis in asparagine-deprived N0 cells. Reduced (**g**) citrate, (**h**) alpha-ketoglutarate and (**i**) malate levels in asparagine-starved A-N0 compared to control C-N5 and W-N5 cells further support (**j**) redirection of TCA cycle flux in asparagine-starved A0 cells. (**c**–**j**) Supplementation with excess asparagine (A-N100) partially reverted aspartate/glutamine ratios and aspartate, glutamate, glutamine, citrate, alpha-ketoglutarate and malate content to levels similar to those in control C-N5 and W-N5 cells. The transcriptome of asparagine-depleted A-N0 and A-N5 mouse sarcoma cells was evaluated by RNA-Seq and compared to A-N100 and C-N5 cells. (**k**) Pathway analyses using the Kyoto Encyclopedia of Genes and Genomes (KEGG) database demonstrated that one of the three top pathways enriched among transcripts upregulated in asparagine-depleted cells was oxidative phosphorylation (false discovery rate (FDR) < 0.05). Higher transcript levels of the (**l**) alpha-ketoglutarate dehydrogenase subunit OGDHL and the (**m**) succinate dehydrogenase subunit SDHB in asparagine-starved A-N0 compared to A-N100 and control C-N5 cells were confirmed by PCR. Please see Appendix A for transcripts upregulated in asparagine-depleted cells and involved in aminoacyl transfer RNA (tRNA) biosynthesis and ribosome. Please see Appendix A for lists of metabolites detected in cell lysates by GC-MS, Appendix A for transcripts upregulated (FDR < 0.05) in A-N0 and A-N5 versus A-N100 and C-N5 cells and Appendix A for pathways enriched (FDR<0.05) in A-N0 and N-N5 cells compared to A-N100 and C-N5 cells. Data were evaluated for statistical significance by one-way ANOVA with Tukey’s post-hoc test (ns *p* ≥ 0.05, * *p* < 0.05, ** *p* < 0.01, *** *p* < 0.001, **** *p* < 0.0001).

**Figure 4 cancers-13-00412-f004:**
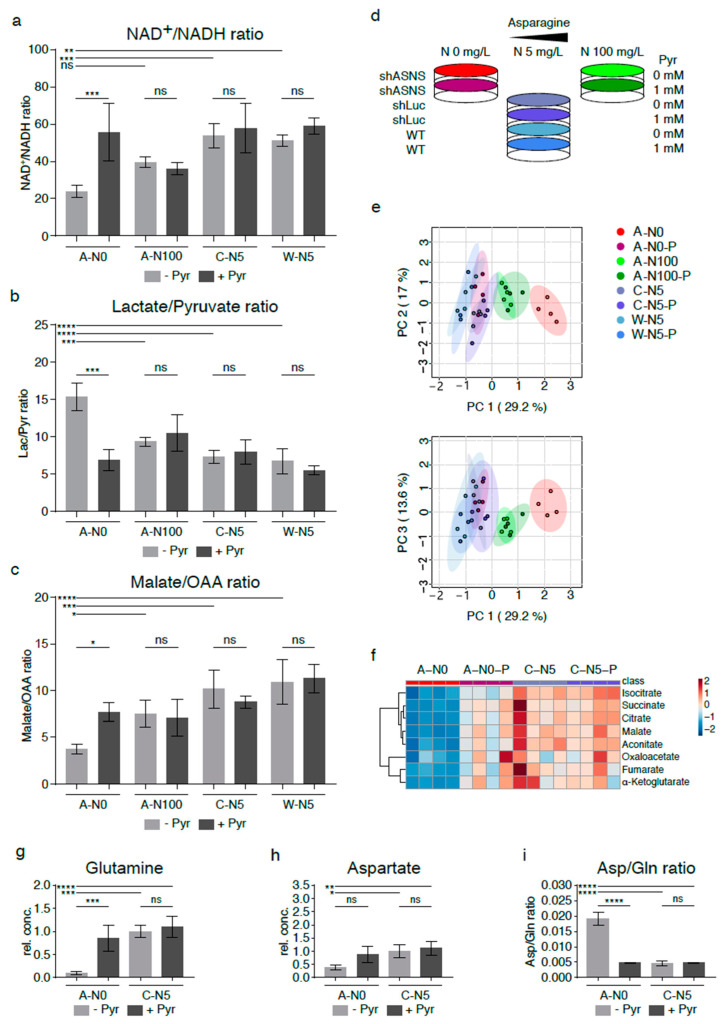
Reversal of altered metabolic flux/reductive stress in asparagine-depleted mouse sarcoma cells by supplementation with pyruvate. (**a**) Relative lack of electron acceptors as evidenced by reduced NAD^+^/NADH rations was consistent with reductive stress in asparagine-starved A-N0 cells compared to control cells cultured in near-physiological asparagine conditions (C-N5, W-N5). Pyruvate and asparagine supplementation rescued the reduced NAD^+^/NADH ratios in asparagine-depleted sarcoma cells. Pyruvate and asparagine supplementation also reversed the (**b**) increased lactate/pyruvate ratios and (**c**) reduced malate/oxaloacetate ratios in A-N0 compared to C-N5/W-N5 cells. (**d**) Asparagine-deprived shASNS cells (A-N0) were grown in medium containing 0mg/L asparagine, 5 mg/L asparagine and 100 mg/L asparagine and supplemented with exogenous pyruvate at a concentration of 88.06 mg/L (1 mM). (**e**) GC-MS data were evaluated by principal component analysis, which demonstrated reversal of the global metabolomic changes in asparagine-depleted cells (A-N0, marked in red) compared to control cells (marked in shades of blue) after pyruvate supplementation (A-N0-P, marked in pink); shASNS cells grown in medium with excess asparagine are marked in green. (**f**) Further LC analyses again demonstrated that reversal of metabolic changes in asparagine-depleted cells (A-N0) included TCA cycle metabolites, which were restored in asparagine-depleted cells supplemented with pyruvate (A-N0-P) to levels comparable to those in control cells (C-N5). (**g**) Pyruvate supplementation restored glutamine levels in asparagine-depleted cells compared to control cells. (**h**) Pyruvate supplementation raised aspartate levels in asparagine-deprived cells, but these changes did not reach significance. (**i**) Pyruvate supplementation restored aspartate/glutamine rations in asparagine-depleted cells. Please see Appendix A for lists of metabolites detected in cell lysates by GC-MS. Please see Appendix A for lower oxygen consumption rates in asparagine-starved compared to control cells. Data were evaluated for statistical significance by one-way ANOVA with Tukey’s post-hoc test (ns *p* ≥ 0.05, * *p* < 0.05, ** *p* < 0.01, *** *p* < 0.001, **** *p* < 0.0001).

**Figure 5 cancers-13-00412-f005:**
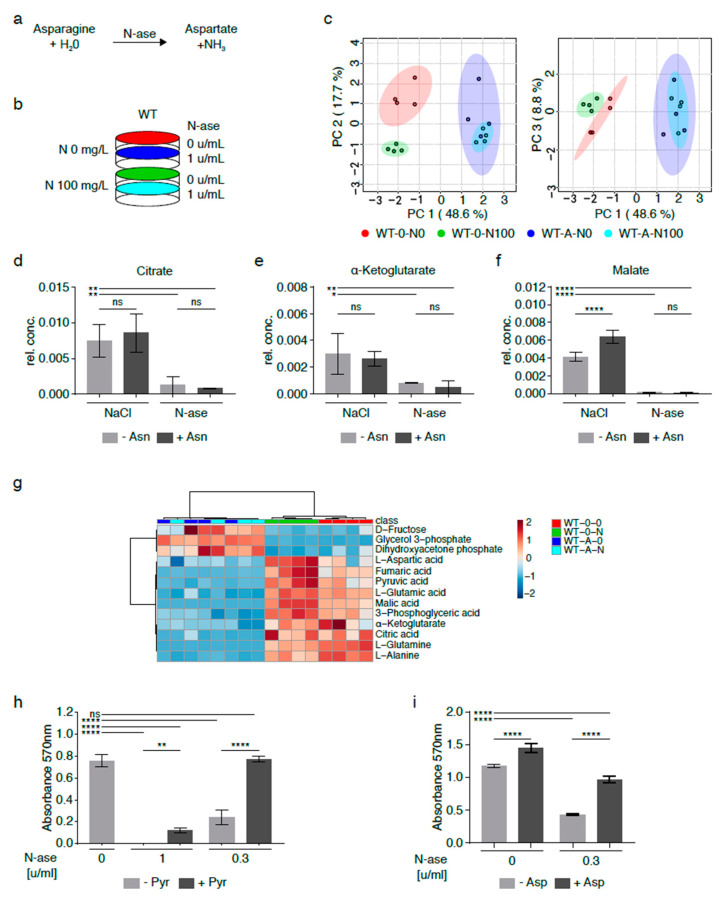
Metabolic adaptation of mouse sarcoma cells treated with asparaginase. (**a**) Asparaginase (N-ase) is an FDA-approved drug which catalyzes the breakdown of asparagine. (**b**) Cells were treated with asparaginase (W-A) or vehicle (W-0) in the presence or absence of supplemental asparagine at a concentration of 100 mg/L. (**c**) The global metabolome of asparaginase-treated sarcoma cells grown with exogenous asparagine (W-A-N100) was evaluated by GC-MS and clustered with asparaginase-treated cells cultured in asparagine-free medium (W-A-N0), but there were marked differences between asparaginase-treated (W-A-N0, W-A-N100) and control sarcoma (W-0-N0, W-0-N100) cells. (**d**–**g**) Alterations in metabolic flux included (**d**) lower citrate, (**e**) lower alpha-ketoglutarate and (**f**) lower malate levels in asparaginase-treated (W-A-N0, W-A-N100) compared to vehicle-treated control (W-0-N0, W-0-N100) cells. (**g**) Overall, the changes observed in asparaginase-treated cells recapitulated the metabolic changes observed in asparagine-deprived shASNS cells cultured in asparagine-free medium compared to control cells. (**h**–**i**) The growth-inhibitory effects of asparaginase on mouse sarcoma cells were reversed by supplementation with (**h**) pyruvate (88.06 mg/L) and (**i**) aspartate (2.66 g/L)). Please see Appendix A for lists of metabolites detected in cell lysates by GC-MS. Data were evaluated for statistical significance by one-way ANOVA with Tukey’s post-hoc test. Data were evaluated for statistical significance by one-way ANOVA with Tukey’s post-hoc test (ns *p* ≥ 0.05, * *p* < 0.05, ** *p* < 0.01, **** *p* < 0.0001).

**Figure 6 cancers-13-00412-f006:**
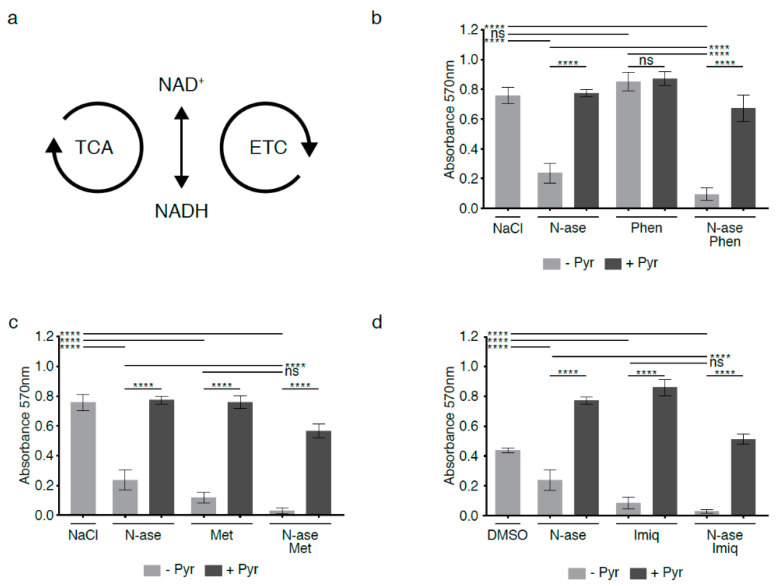
Synergistic growth-inhibitory effects of complex 1 inhibitors and asparaginase on mouse sarcoma cells. (**a**) Flux through the tricarboxylic acid cycle (TCA) and electron transport chain (ETC) is associated with a continuous flow of electrons, during which NAD^+^ serves as an electron carrier in continuous cycles of reduction to NADH (e.g., via the TCA cycle) and oxidation back to NAD^+^ (e.g., via the electron transport chain (ETC)). (**b**) Mouse sarcoma cells were exposed to low concentrations of the complex 1 inhibitor phenformin, which did not reduce sarcoma cell proliferation. However, combinatorial exposure to phenformin and asparaginase augmented the growth-inhibitory effects of asparaginase alone. Supplementation with the exogenous electron acceptor pyruvate reversed the growth inhibitory effects of phenformin and asparaginase, alone and in combination. (**c**,**d**) Two alternative complex 1 inhibitors (metformin and imiquimod) reduced mouse sarcoma cell proliferation and deepened the anti-proliferative effects of asparaginase. Again, the growth-inhibitory effects of asparaginase and metformin/imiquimod alone and in combination were reversed by supplementing cells with pyruvate. Chemicals were added using the following concentrations: phenformin 10 µM, metformin 1 mM, imiquimod 20 µM, asparaginase 0.3 U/mL and pyruvate 88.06 mg/L. Data were evaluated for statistical significance by one-way ANOVA with Tukey’s post-hoc test (ns *p* ≥ 0.05, **** *p* < 0.0001).

**Figure 7 cancers-13-00412-f007:**
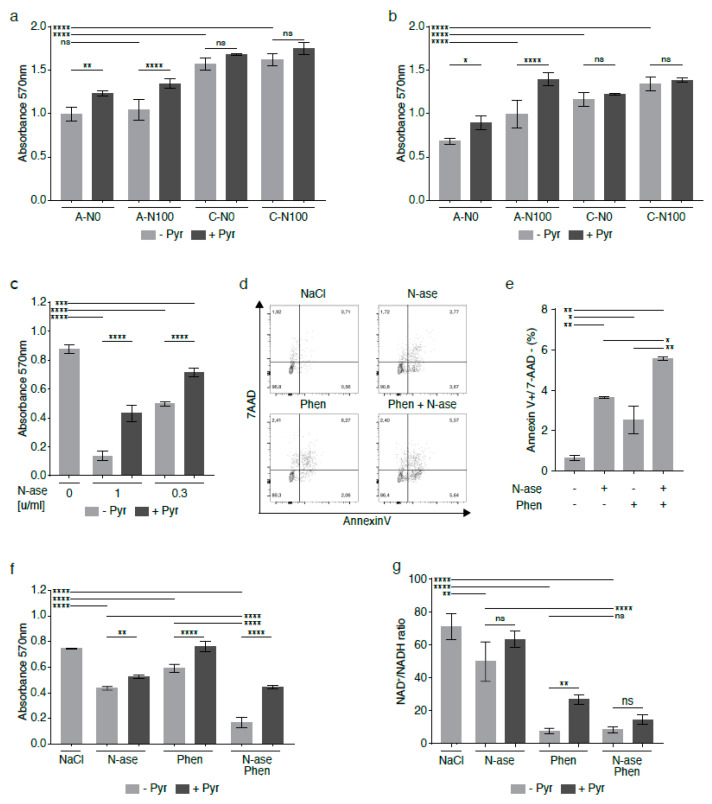
Synergistic growth-inhibitory effects of phenformin and asparaginase on human RD rhabdomyosarcoma cells**.** In the human rhabdomyosarcoma cell line RD, asparagine deprivation by shASNS knockdown and culture in asparagine-free medium reduced proliferation in (**a**) supraphysiological glutamine conditions and in (**b**) physiological glutamine conditions. (**a**,**b**) The anti-proliferative effects of asparagine deprivation were reversed by supplementation with exogenous asparagine in (**b**) a physiological glutamine, but not (**a**) a supraphysiological glutamine, environment. (**a**,**b**) The anti-proliferative effects of asparagine deprivation were reversed by supplementation with exogenous pyruvate in both physiological glutamine and supraphysiological glutamine conditions. (**c**–**e**) Asparaginase, which hydrolyzes both asparagine and glutamine in the cell environment, (**c**) reduced the growth and (**d**,**e**) raised apoptosis of human RD cells at certain concentrations. (**c**) Supplementation with exogenous pyruvate reversed the anti-proliferative effects of asparaginase treatment. (**d**,**e**) Combinatorial treatment with asparaginase and phenformin deepened the pro-apoptotic effects of asparaginase on RD cells. (**f**) Phenformin also augmented the growth-inhibitory effects of asparaginase on RD cells. The anti-proliferative effects of phenformin and asparaginase, alone and in combination, were again reversed by addition of pyruvate. (**g**) NAD^+^/NADH ratios were reduced in RD cells treated with asparaginase and phenformin alone and in combination. Supplementation with pyruvate partially restored NAD^+^/NADH ratios in RD cells treated with phenformin alone. Pyruvate also raised the NAD^+^/NADH ratios in RD cells treated with asparaginase alone or asparaginase and phenformin in combination, but these changes did not reach statistical significance. Chemicals were added using the following concentrations for proliferation assays: phenformin 10 µM, asparaginase 0.3 U/mL and pyruvate 88.06 mg/L. Chemicals were added using the following concentrations for apoptosis assays: phenformin 50 µM and asparaginase 1 U/mL. Please see Appendix A for metformin and imiquimod effects on RD cells, alone and in combination with asparaginase. Data were evaluated for statistical significance by one-way ANOVA with Tukey’s post-hoc test (ns *p* ≥ 0.05, * *p* < 0.05, ** *p* < 0.01, *** *p* < 0.001, **** *p* < 0.0001).

**Figure 8 cancers-13-00412-f008:**
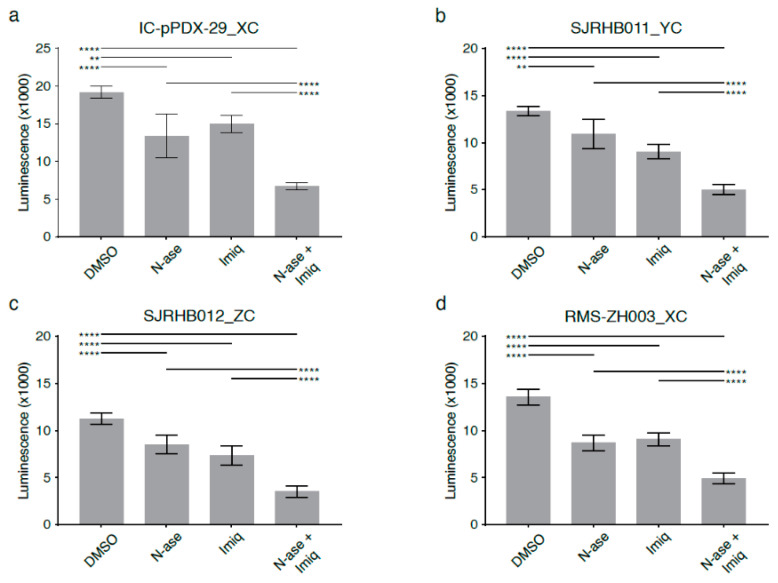
Synergistic growth-inhibitory effects of phenformin and asparaginase on human primary rhabdomyosarcoma cell cultures. Asparaginase and the complex 1 inhibitor imiquimod reduce the proliferation of primary patient-derived human rhabdomyosarcoma cultures (**a**) Ic pPDX-29_XC, (**b**) SJRHB011_YC, (**c**) SJRHB012_ZC and (**d**) RMS-ZH003_XC. (**a**–**d**) Simultaneous exposure to asparaginase and imiquimod enhances the anti-proliferative effects of both compounds in all four primary rhabdomyosarcoma cell cultures. Data were evaluated for statistical significance by one-way ANOVA with Tukey’s post-hoc test (** *p* < 0.01, **** *p* < 0.0001).

## Data Availability

Metabolomics raw data are accessible on the metabolights repository (https://www.ebi.ac.uk/metabolights/MTBLS2035). RNA-Seq raw data are accessible on GEO (GSE153991; token: ozgdacsyhihnkf).

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
