# Peer review of "Lack of Electron Acceptors Contributes to Redox Stress and Growth Arrest in Asparagine-Starved Sarcoma Cells"

_cancers, 2021, doi:10.3390/cancers13030412_

Round 1

Reviewer 1 Report

In this manuscript, Bauer et al. investigate the consequences of asparagine starvation in sarcoma cells with altered asparagine synthesis capacity. Consistent with dependency on asparagine for cell proliferation, sarcoma cells expressing an shRNA that depletes ASNS expression have impaired proliferation upon asparagine withdrawal that can be restored with exogenous asparagine. Metabolomic profiling shows that asparagine starvation by shASNS and asparagine withdrawal leads to depletion of TCA cycle intermediates and related amino acids and increases expression of oxidative phosphorylation genes. Asparagine starvation also causes impairments in NAD+/NADH homeostasis, which can be mitigated by treatment with an exogenous electron acceptor, leading to restoration of some metabolic parameters. Asparagine depletion by treatment with the chemotherapeutic enzyme L-asparaginase can also similarly deplete metabolite levels and cell proliferation. Consistent with a role of redox stress, the authors show that asparagine depletion by L-asparaginase treatment can enhance the antiproliferative effects of mitochondrial inhibition across several sarcoma cell lines.

Overall, the manuscript is broadly consistent with the current understanding of the role of mitochondrial metabolism in supporting aspartate and asparagine synthesis, however this work expands this observation to several sarcoma cell lines. I have only a couple recommended experiments:

In figure 2c the authors show intracellular asparagine levels in mouse sarcoma cells in shASNS-0 asparagine, shluciferase-5asparagine, and WTcontrol-5asparagine. Having two variables altered at the same time makes it difficult to understand if shASNS or the lack of extracellular asparagine are the primary reason that asparagine is depleted in the first condition. Can the authors repeat across all 3 cell lines at both 0 and 5 mg/L asparagine?

In Figure 5, the authors use L-asparaginase to deplete asparagine levels in the media. One commonly described off target activity of L-asparaginase is that it can also function as a glutaminase, albeit with lower activity. Since the authors also observe glutamine depletion in cells upon L-asparaginase treatment, it would seem worthwhile to determine if the doses of L-asparaginase they are using also deplete glutamine in the media. This is one possible explanation for why pyruvate is able to rescue cell proliferation from 0.3 U/ml L-asparaginase, but not 1 U/ml.

Author Response

Author’s reply to reviewer #1:

In this manuscript, Bauer et al. investigate the consequences of asparagine starvation in sarcoma cells with altered asparagine synthesis capacity. Consistent with dependency on asparagine for cell proliferation, sarcoma cells expressing an shRNA that depletes ASNS expression have impaired proliferation upon asparagine withdrawal that can be restored with exogenous asparagine. Metabolomic profiling shows that asparagine starvation by shASNS and asparagine withdrawal leads to depletion of TCA cycle intermediates and related amino acids and increases expression of oxidative phosphorylation genes. Asparagine starvation also causes impairments in NAD+/NADH homeostasis, which can be mitigated by treatment with an exogenous electron acceptor, leading to restoration of some metabolic parameters. Asparagine depletion by treatment with the chemotherapeutic enzyme L-asparaginase can also similarly deplete metabolite levels and cell proliferation. Consistent with a role of redox stress, the authors show that asparagine depletion by L-asparaginase treatment can enhance the antiproliferative effects of mitochondrial inhibition across several sarcoma cell lines.

Overall, the manuscript is broadly consistent with the current understanding of the role of mitochondrial metabolism in supporting aspartate and asparagine synthesis, however this work expands this observation to several sarcoma cell lines. I have only a couple recommended experiments:

We appreciate the reviewer’s thorough review of our manuscript and positive feedback.

In figure 2c the authors show intracellular asparagine levels in mouse sarcoma cells in shASNS-0 asparagine, shluciferase-5asparagine, and WTcontrol-5asparagine. Having two variables altered at the same time makes it difficult to understand if shASNS or the lack of extracellular asparagine are the primary reason that asparagine is depleted in the first condition. Can the authors repeat across all 3 cell lines at both 0 and 5 mg/L asparagine?

Ad Figure 2c) The revised manuscript contains LC-MS data on relative asparagine content in shASNS, shLuc and wild-type cells grown in medium with 0 mg/L asparagine, 5 mg/L asparagine and 100 mg/L asparagine. Supplementation with 5mg/mL asparagine did not reverse the effects of the shASNS knockdown. Supplementation with 100mg/ml asparagine raised asparagine content in shASNS, shLuc and wild-type cells to supraphysiological levels (please see revised figure S1 and revised manuscript, page 8, 2nd paragraph).

In Figure 5, the authors use L-asparaginase to deplete asparagine levels in the media. One commonly described off target activity of L-asparaginase is that it can also function as a glutaminase, albeit with lower activity. Since the authors also observe glutamine depletion in cells upon L-asparaginase treatment, it would seem worthwhile to determine if the doses of L-asparaginase they are using also deplete glutamine in the media. This is one possible explanation for why pyruvate is able to rescue cell proliferation from 0.3 U/ml L-asparaginase, but not 1 U/ml.

Ad Figure 5) We appreciate the reviewer’s important comment and respectfully point out that Purwaha et al investigated glutamine content in medium (with and without cells) during treatment with vehicle, 0.1 U/ml and 0.5 U/ml asparaginase using LC-MS. They observed a decline in glutamine levels in medium over 48 hours (Purwaha et al, Metabolomics, 2014, PMID 25177232). Glutamine levels were similar after 48 hours of treatment with 0.5 U/ml and 0.1 U/ml asparaginase. We commented on glutamine content in medium exposed to asparaginase in the revised manuscript (please see revised manuscript, page 19 first paragraph).

All changes in the revised manuscript were underscored.

Reviewer 2 Report

Dear Editors:

Cancer cells require substantial supply of nutrients to support their fast growth, so that they are hypersensitive to insufficient availability of not only essential, but also some of non-essential amino acids. Depletion of intracellular asparagine through bacterial derived asparaginase has long been a major component of leukemia therapy. Bauer et.al has observed that genetic silencing of ASNS combined with restricted supply of asparagine decreased sarcoma cell growth. In this report, with both mouse and human sarcoma cells they further demonstrated that: 1). Asparagine depletion redirect the TCA flux to increase synthesis of aspartate accompanying the upregulation of enzymes involving these pathways, such as oxidative phosphorylation. 2). Redirection of TCA flux leads to reductive stress by reducing amount of NAD+ electron carriers. 3).These changes can be reversed by supplementation with pyruvate. 4). lastly and most importatnly, complex I inhibitor treatment augmented reductive stress and growth inhibiton induced by asparagine deprivation. The study is sound and solid, and it reveals for the first time the molecular mechanism by which a non-essential amino acid affect cancer cell growth and survival. It lays a foundation for further exporation for a potential inhibitor to cure cancers. I strongly recommend for publication.

  1. Line 67, “5.4 mg/L (82.4 uM)” is incorrect.
  2. Line 238, “Aminotransferase” not “Amidotransferase”

Author Response

Author’s reply to reviewer #2:

We are grateful for the favorable feedback on our manuscript.

Ad asparagine concentration in human serum) Psychogios et al (PLoS One, 2011, PMID 21359215) determined that the concentration asparagine in human serum (obtained from healthy individuals) was 82.4 +/- 7.3 mM. This equals 10.9 mg/L (not 5.4 mg/L). This was changed in the revised manuscript. We apologize for the error.

Ad designation “amidotransferase”) The designation was changed from “amidotransferase” to “aminotransferase”.

All changes in the revised manuscript were underscored.

Round 2

Reviewer 1 Report

The authors have satisfactorily addressed my critiques.